# Ecological adaptation in Atlantic herring is associated with large shifts in allele frequencies at hundreds of loci

Fan Han[1], Minal Jamsandekar[2], Mats E Pettersson[1], Leyi Su[1], Angela P Fuentes-Pardo[1], Brian W Davis[2], Dorte Bekkevold[3], Florian Berg[4,5], Michele Casini[6,7], Geir Dahle[5], Edward D Farrell[8], Arild Folkvord[4,5], Leif Andersson[1,2,9]*

[1]Department of Medical Biochemistry and Microbiology, Uppsala University, Uppsala, Sweden; [2]Department of Veterinary Integrative Biosciences, Texas A&M University, College Station, United States; [3]National Institute of Aquatic Resources, Technical University of Denmark, Silkeborg, Denmark; [4]Department of Biological Sciences, University of Bergen, Bergen, Norway; [5]Institute of Marine Research, Bergen, Norway; [6]Department of Aquatic Resources, Institute of Marine Research, Swedish University of Agricultural Sciences, Lysekil, Sweden; [7]Department of Biological, Geological and Environmental Sciences, University of Bologna, Bologna, Italy; [8]School of Biology and Environmental Science, Science Centre West, University College Dublin, Dublin, Ireland; [9]Department of Animal Breeding and Genetics, Swedish University of Agricultural Sciences, Uppsala, Sweden

**Abstract** Atlantic herring is widespread in North Atlantic and adjacent waters and is one of the most abundant vertebrates on earth. This species is well suited to explore genetic adaptation due to minute genetic differentiation at selectively neutral loci. Here, we report hundreds of loci underlying ecological adaptation to different geographic areas and spawning conditions. Four of these represent megabase inversions confirmed by long read sequencing. The genetic architecture underlying ecological adaptation in herring deviates from expectation under a classical infinitesimal model for complex traits because of large shifts in allele frequencies at hundreds of loci under selection.

*For correspondence:
leif.andersson@imbim.uu.se

**Competing interests:** The authors declare that no competing interests exist.

## Introduction

Atlantic herring (*Clupea harengus*) constitutes the basis for one of the world's most important commercial fisheries (*Food and Agriculture Organization of the United Nations Statistics Division, 2013*) and has been a valuable food resource throughout human history in North Europe. It is one of the most abundant vertebrates on earth with a total estimated breeding stock of one trillion individuals (*Feng et al., 2017*). Its short-term (<10,000 years) effective population size is enormous whereas the long-term effective population size is much more restricted most likely due to the impact of periods of glaciation (*Martinez Barrio et al., 2016*). It is likely that genetic draft (*Walsh and Lynch, 2018*) that is purging of neutral genetic variation linked to polymorphisms under positive or negative selection, also contributes to restricting nucleotide diversity in the herring. As a consequence, the nucleotide diversity is moderate (~0.3%) (*Martinez Barrio et al., 2016*), only three-fold higher than in human although the census population size is at least two orders of magnitude higher in herring at present and until recently (>1,000 years ago) many orders of magnitude higher.

Detailed population genomic studies of Atlantic herring are justified for two reasons. Firstly, a better understanding of the population structure and development of diagnostic markers have the potential to revolutionize the procedures of stock assessments of herring that has been in operation for more than hundred years, which will add key scientific knowledge to safeguard a sustainable fishery. Secondly, the Atlantic herring can be used as a model to study the genetic architecture underlying phenotypic diversity and ecological adaptation. The herring has adapted to different marine environments including the brackish Baltic Sea, where salinity drops to 2–3 PSU and water temperature shows a much larger variation among seasons than in the Atlantic Ocean with typically 34–35 PSU (*Figure 1—figure supplement 1*). Furthermore, herring spawn during different periods of the year, which involves photoperiodic regulation of reproduction. There is minute genetic differentiation at selectively neutral loci among herring populations from different geographic regions (*Lamichhaney et al., 2012*; *Martinez Barrio et al., 2016*) and the distribution of per locus $F_{ST}$ values deviates significantly from the one expected for selectively neutral loci with an excessive number of outlier loci demonstrating signatures of selection (*Lamichhaney et al., 2017*; *Limborg et al., 2012*). These properties make the Atlantic herring a powerful model to explore how natural selection shapes the genome in a species where genetic drift plays a minor role in genome-wide population divergence.

Understanding the genetic basis for complex traits and disorders is of central importance in current biology and human medicine. The aim of the present study was to explore the genetic architecture underlying ecological adaptation in herring by a comprehensive analysis of whole-genome sequence data from 53 population samples spread across the entire species distribution.

## Results

### Genome sequencing

We performed pooled whole-genome sequencing using 28 population samples of Atlantic herring. These were analyzed together with previously published data (*Lamichhaney et al., 2017*; *Martinez Barrio et al., 2016*), making a total number of 53 population samples. The current study has added populations around Ireland and Britain, from Norwegian fjords and from the southern Baltic Sea. Together the 53 population samples cover the entire species distribution and include spring, autumn, winter and summer spawning populations (*Figure 1* and *Supplementary file 1*). A previously sequenced population of Pacific herring (*Clupea pallasii*) was used as a closely related outgroup in the bioinformatic analysis. We pooled genomic DNA from 35 to 110 individuals per population, and conducted whole-genome resequencing to a minimum of 30 × coverage per population. In order to generate individual genotype data, we sequenced 12 fish from four localities to ~10 × coverage per individual. Together with data from previous studies (*Lamichhaney et al., 2017*; *Lamichhaney et al., 2012*; *Martinez Barrio et al., 2016*), we used a total of 55 individually sequenced herring in the analysis (*Supplementary file 2*). All sequence reads were aligned to the recently released chromosome-level assembly for Atlantic herring (*Pettersson et al., 2019*), and sequence variants were called and filtered with a stringent pipeline (Methods). In total, we identified ~11.5 million polymorphic biallelic sites among the 53 population samples and ~15.9 million sites when including Pacific herring.

### Detection of population structure

A genetic distance tree, based on genome-wide SNPs, groups the 53 population samples into seven primary clusters: (i) autumn- and (ii) spring-spawning herring from the brackish Baltic Sea, (iii) populations from the transition zone, close to the entrance to the Baltic Sea, spawning at lower salinity than in the Atlantic Ocean, (iv) Norwegian fjord populations, (v) populations around Ireland and Britain, (vi) autumn- and (vii) spring-spawning herring from the North Atlantic Ocean (*Figure 2—figure supplement 1*). This clustering, in general, fits with their geographical origin, but the branch lengths separating some subpopulations are very short, for example autumn- and spring-spawning populations from the North Atlantic Ocean. Pairwise $F_{ST}$ values among all populations are in the range 0.013 to 0.061 (*Figure 2—figure supplement 2*).

To explore the contribution of strongly differentiated loci to the observed population structure, we separated SNPs showing no significant genetic differentiation between populations from those

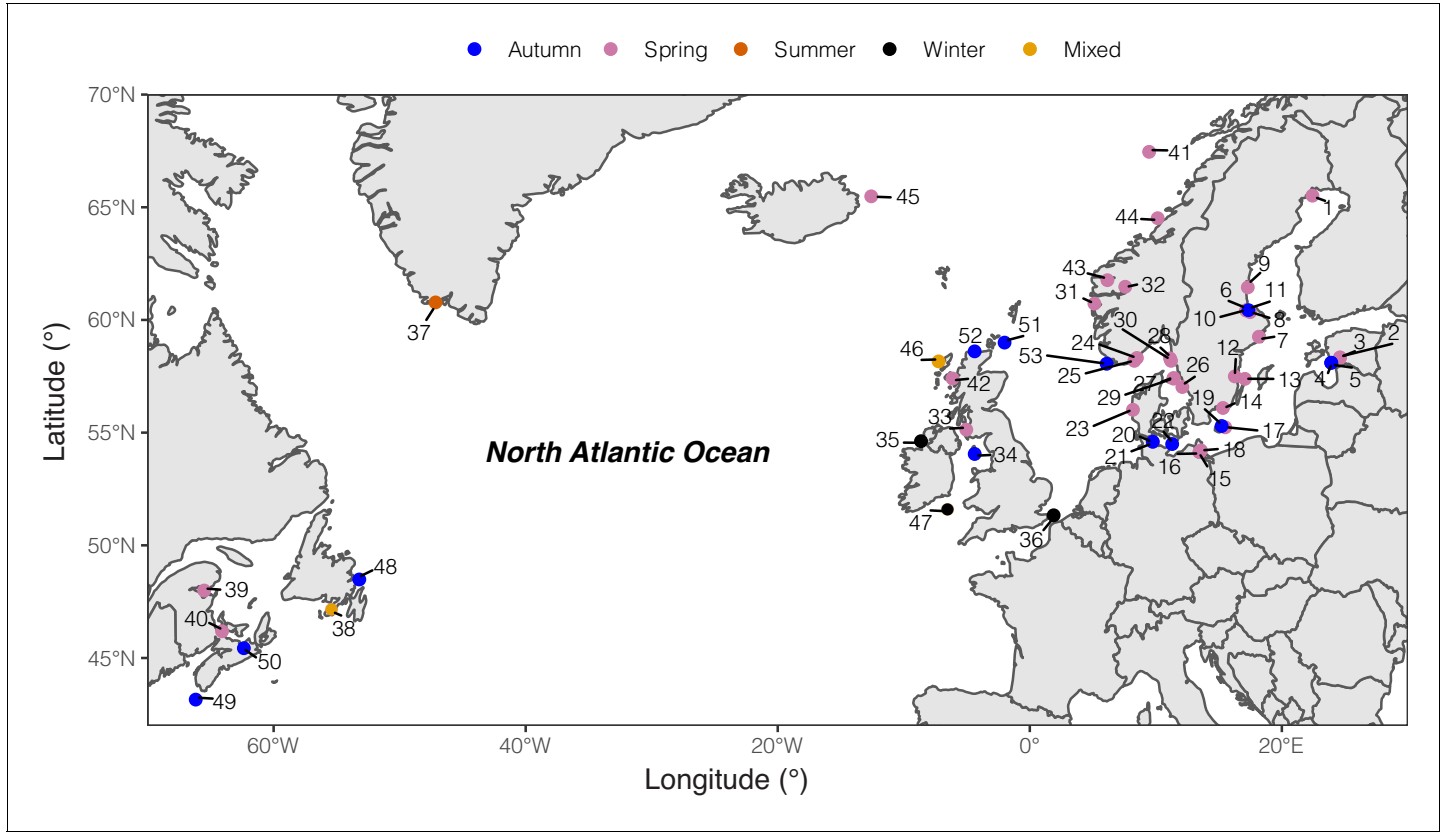

**Figure 1.** Geographic distribution of sampled pools of Atlantic herring. Color code of each population refers to the spawning season that a given pool was sampled.

The online version of this article includes the following figure supplement(s) for figure 1:

**Figure supplement 1.** Variation in sea water temperatures among seasons.

showing strong genetic differentiation based on their standard deviation of allele frequencies across Atlantic and Baltic populations (Methods; *Figure 2—figure supplement 3*). We carried out Principal Component Analysis (PCA) based on these two sets of markers independently. The PCA using 169,394 undifferentiated markers separated some of the major groups but three distinct groups (Atlantic Ocean spring and autumn-spawners, and populations around Ireland and Britain) were merged into one tight cluster (*Figure 2a*). In fact, the populations within this very tight cluster constitute ~90% of the world population of Atlantic herring (see *Feng et al., 2017*, *Supplementary file 1*), all spawning in full marine environment (salinity ~35 PSU). The first two principal components in this analysis explained only ~11% of the variance. This is in line with the conclusion that the majority of the genome shows minute genetic differentiation across the entire species distribution of Atlantic herring (*Fuentes-Pardo et al., 2019*; *Lamichhaney et al., 2017*; *Lamichhaney et al., 2012*; *Martinez Barrio et al., 2016*). In contrast, the PCA based on only 794 markers showing the most striking genetic differentiation separated the populations into the seven main clusters (*Figure 2b*), consistent with the genetic distance tree (*Figure 2—figure supplement 1*). The first principal component explained 43% of the variance. These two patterns observed for the two sets of markers suggest that the observed population structure in herring can be accounted for by a small fraction of genetic markers associated with loci underlying ecological adaptation.

The PCA analysis based on the differentiated SNPs revealed some intriguing results as regards population classifications (*Figure 2b*). Two population samples (#24 and 25) both from Landvikvannet in southern Norway clustered more closely with populations from the transition zone than with local fjord populations from western Norway. Landvikvannet is a unique brackish lake created in 1877 by opening a canal between the ocean and a freshwater lake, now harboring a local population of herring adapted to this brackish environment (*Eggers et al., 2014*). The populations around

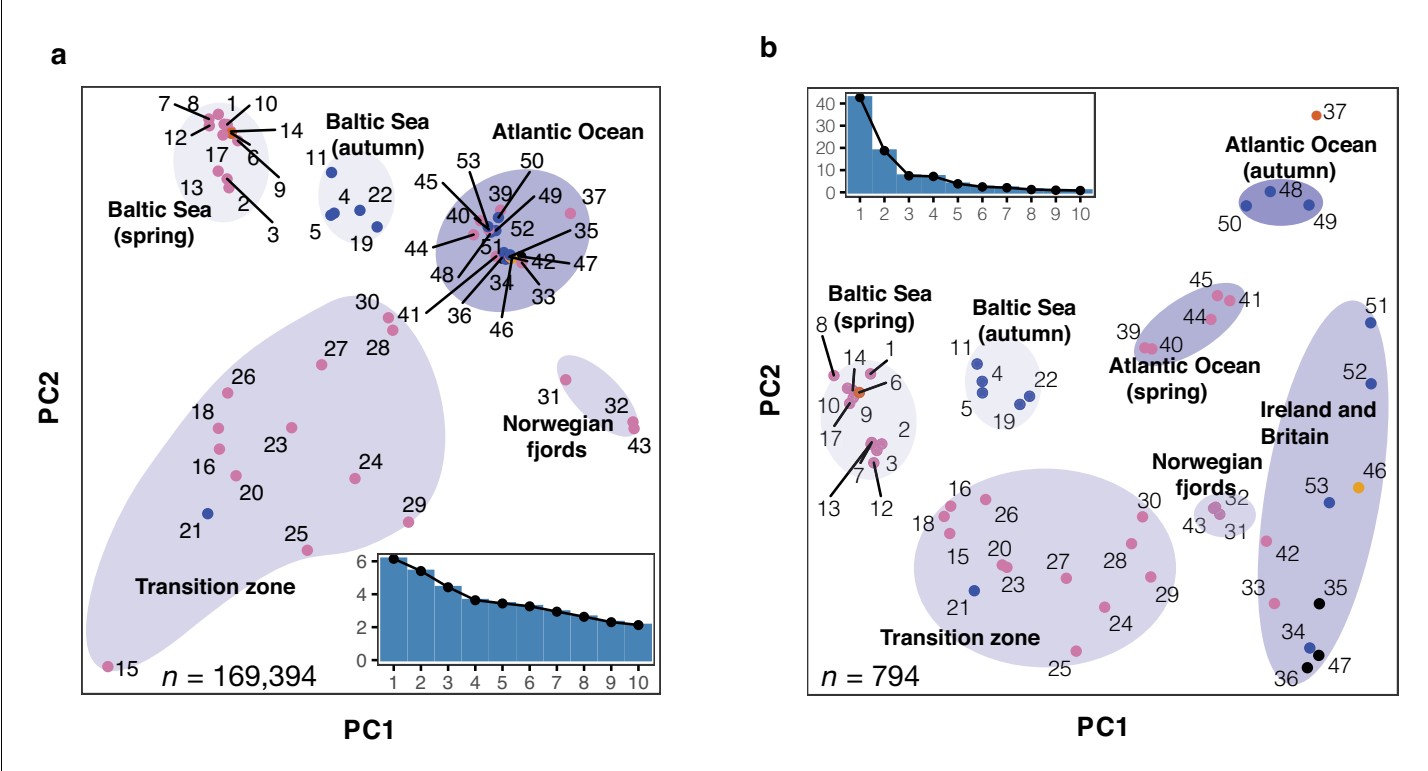

**Figure 2.** Principal component analysis (PCA) of herring populations. (a) PCA based on 169,934 genetically undifferentiated markers. (b) PCA based on 794 markers that are likely under natural selection. Color of each population represents spawning season, as coded in *Figure 1*. Inset bar plot indicates the percentage of explained variance by each principal component. n = number of SNPs used in the analysis.

The online version of this article includes the following figure supplement(s) for figure 2:

**Figure supplement 1.** Phylogenetic reconstruction.

**Figure supplement 2.** Pairwise $F_{ST}$ among 53 populations of Atlantic herring.

**Figure supplement 3.** Histogram of genome-wide standard deviation of allele frequencies across 52 Atlantic and Baltic populations.

Ireland and Britain showed a relatively high degree of diversity, despite their geographical proximity. Populations within this cluster do not show such distinct sub-clusters according to spawning season as they do for instance in the Baltic Sea. One of the populations from Schlei (#21), collected as autumn-spawning, clustered with spring-spawning populations. Interestingly, examinations of the otoliths from fish in this sample, using the method by *Clausen et al., 2007*, showed that they hatched in spring! Thus, this must represent a group of herring that has switched spawning season from spring to autumn, maybe because their nutritional status was insufficient to support spawning (*McQuinn, 1997*), demonstrating that spawning time in herring is controlled by a combination of genetic and environmental factors (*Bekkevold et al., 2007*; *Berg et al., 2020*).

Taken together, in Atlantic herring, a modest number of genetic markers showing strong genetic differentiation provide a much better resolution to distinguish population structure than a large number of random neutral genetic markers.

## Genetic signatures of ecological adaptation

To explore the genetic architecture underlying ecological adaptation in the herring, we formed superpools of populations representing the major groups detected using the PCA analysis (*Figure 2b*), and performed four genome-wide contrasts (*Supplementary file 3*), (i) Baltic herring vs. Northeast Atlantic herring, (ii) spring-spawning vs. autumn-spawning herring, (iii) Atlantic herring from the waters around Ireland and Britain vs. other parts of the Northeast Atlantic Ocean, and (iv) Atlantic herring from the North Atlantic Ocean vs. all other Atlantic herring samples but excluding herring from the Baltic Sea and Norwegian fjords (*Figure 2b*). In each contrast, we compared allele

frequency differentiation on a per SNP basis using a $\chi^2$ test, corrected for inflation due to baseline differences, and used a stringent significance threshold ($p < 1 \times 10^{-10}$, Methods). While exact numbers varied due to sample composition, all contrasts involved approximately six million SNPs.

## Baltic herring vs. northeast Atlantic herring

We divided this contrast into two replicates using the genetically distinct spring- and autumn-spawning populations from each region. The Baltic/Northeast Atlantic contrast using 19 spring-spawning and 9 autumn-spawning populations resulted in a total number of 115 and 39 loci, respectively,

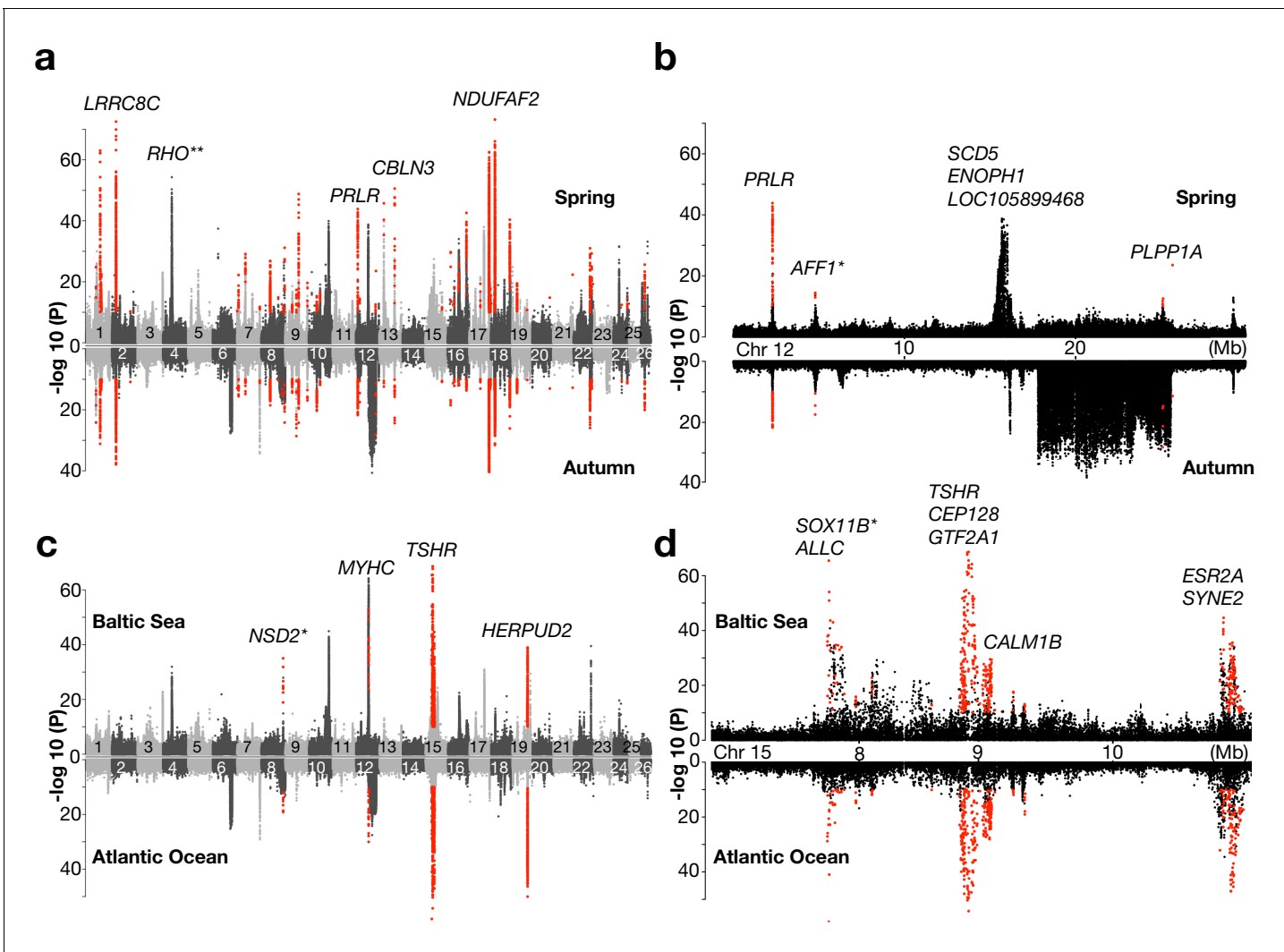

**Figure 3.** Genetic signals associated with ecological adaptation. (**a**) Genetic differentiation between superpools of Baltic and Atlantic herring in spring-spawners (above) and in autumn-spawners (below) and (**b**) a zoomed-in profile for Chr12. (**c**) Genetic differentiation between super-pools of spring and autumn herring in Baltic (above) and in Atlantic herring (below) and (**d**) a zoomed-in profile on Chr15. Red dots indicate signals shared between the two contrasts. *Peak is located in intergenic region; the closest gene is labeled. **An identified locus associated with adaptation to the red-shifted light environment in the Baltic Sea.

The online version of this article includes the following figure supplement(s) for figure 3:

**Figure supplement 1.** Genetic differentiation between superpools of Atlantic and Baltic herring in spring-spawners (above the x-axis) and autumn-spawners (below the x-axis).

**Figure supplement 2.** Genetic differentiation around (**a**) *PRLR* gene and (**b**) *SOX11B* gene in each contrast.

**Figure supplement 3.** Genetic differentiation between super-pools of spring and autumn-spawning herring in Atlantic (above) and in Baltic herring (below).

**Figure supplement 4.** Comparisons of p-values and delta allele frequencies (DAF) for the contrasts shown in *Figure 3*.

reaching statistical significance (*Figure 3a*; results for individual chromosomes are in *Figure 3—figure supplement 1* and all highly differentiated SNPs are listed in *Supplementary file 4*). As many as 30 of these loci reached significance in both comparisons (highlighted in red in *Figure 3a*). These loci are the best candidates for adaptation to the environmental conditions in the brackish Baltic Sea irrespective of spawning time. A prime example of such a locus is *LRRC8C* (*leucine-rich repeat-containing 8 protein, subunit C*) ($p<10^{-72}$, in spring-spawners and $p<10^{-37}$ in autumn-spawners) which encodes a volume-regulated anion-channel. Another locus that was highly significant in both replicate comparisons is *PRLR* (*prolactin receptor*) on Chr12 highlighted in *Figure 3b*. The most significant association at this locus is a cluster of non-coding SNPs located in a gene desert upstream of *PRLR* (*Figure 3—figure supplement 2a*). This hormone-receptor has an important role during lactation in mammals and a critical role for osmoregulation and adaptation to salinity in fish (*Manzon, 2002*). The loci that do not replicate in the two subsets are not false positives but are less likely to be directly related to adaptation to brackish water. A striking example of such a locus is the large 7.8 Mb region on Chr12 that shows no genetic differentiation in the spring-spawning contrast, but very strong genetic differentiation between autumn-spawning Atlantic and Baltic herring (*Figure 3b*). This region corresponds to a recently reported inversion (*Pettersson et al., 2019*) (see further data below).

## Spring-spawning vs. autumn-spawning herring

We divided this contrast into two replicates: (i) populations from Baltic Sea and (ii) populations from the Northeast Atlantic Ocean. The spring-/autumn-spawning contrast involved 16 Baltic and 12 Northeast Atlantic populations, which yielded 31 and 13 significant loci, respectively (*Figure 3c*; results for individual chromosomes are in *Figure 3—figure supplement 3* and all highly differentiated SNPs are listed in *Supplementary file 5*). Only seven loci were significant in both replicates. Out of the seven replicated loci, three were found within 5 Mb on Chr15 (*Figure 3d*). The most significant signal among those contains the *TSHR* (*Thyroid-Stimulating Hormone-Receptor*) locus ($p<10^{-68}$ in Baltic populations and $p<10^{-53}$ in Atlantic populations), which has a well-defined role in the pathway regulating seasonal reproduction in vertebrates (*Nakane et al., 2013*; *Nakane and Yoshimura, 2019*) and *CALM1B* (*Calmodulin*), which regulates gonadotrophin-releasing hormone signaling via the MAPK pathway (*Melamed et al., 2012*). Another significant locus on Chr15 is *SOX11B* (*SRY-Box Transcription Factor 11*), which plays an essential role in sex determination in zebrafish (*Santos et al., 2007*). Notably, the most significant sequence variant at this locus ($p<10^{-65}$ in Baltic populations and $p<10^{-57}$ in Atlantic populations) is a single-base change in a non-coding region located 39 kb upstream of *SOX11B* (*Figure 3—figure supplement 2b*), suggesting that it may be a causal regulatory mutation. The third highly significant locus on Chr15 contains *ESR2A* (*Estrogen Receptor Beta*) and *SYNE2* (*Spectrin Repeat Containing Nuclear Envelope 2*), and these two genes are only 100 kb apart (*Figure 3d*). In fact, a closer examination of this region shows that there are two distinct signals at the two genes. *ESR2A* is an obvious candidate gene for a reproduction-related phenotype, suggesting that the signal at *SYNE2* may reflect regulatory variants affecting *ESR2A* expression. However, *SYNE2* encodes an important structural protein and a premature stop codon in this gene is reported to cause defects in photoreceptors in mice (*Maddox et al., 2015*), suggesting that it may in fact have a role in photoperiod regulation of reproduction in herring.

In addition to the signals on Chr15, we also found other loci that are associated with adaptation to different spawning seasons, such as multiple members of *MYHC* (myosin heavy chain) gene family on Chr12, which may be related to a reported association between myogenesis and plasticity of seasonal development in herring (*Johnston et al., 2001*). Another locus is *HERPUD2* (*Homocysteine Inducible ER Protein with Ubiquitin-like Domain 2*) on Chr19, a gene that has no known function in reproductive biology.

## Atlantic herring from the waters around Ireland and Britain show distinct signatures of selection

An important geographical area missing in our previous genetic screens are the waters around Ireland and Britain, which are inhabited by the southernmost ecomorphs of herring in the Northeast Atlantic that spawn in warmer sea water than other herring populations (*Figure 1—figure supplement 1*). We grouped 10 populations from this region as one superpool, and compared its genome-

wide allele frequency with another superpool, which consisted of 4 neighboring populations from the Northeast Atlantic Ocean (*Figure 4a*). Following the definition of independent loci (see Methods), we found only 12 independent loci showing significant genetic differentiation in this contrast. Several of these are the loci also detected in the contrast between spring- vs autumn-spawning herring (*TSHR*, *HERPUD2*, etc.), as expected due to the difference in proportion of autumn-spawners (40% vs. 0%). However, four of the major loci on Chr6, 12, 17, and 23 span exceptionally large regions on four different chromosomes (*Figure 4a*). Unlike a typical selective sweep where divergence of the markers forms a bell shape, the zoom-in profile of the significant SNPs at each locus formed a block-like pattern, where a sharp change of divergence was observed at borders (*Figure 4b–e*). The block-like patterns, suggesting suppressed recombination, are consistent with the presence of inversions or other structural rearrangements. The very sharp borders of these regions are expected for inversions maintained as balanced polymorphisms, because recombination over thousands of generations will randomize the association between the inversion haplotypes and flanking markers. A classic example of this pattern is the rapid decay of linkage disequilibrium in the region flanking the ~4 million year old inversion underlying alternative male mating strategies in the ruff (*Lamichhaney et al., 2016*).

Further exploration of allele frequencies at diagnostic markers at each putative structural variant showed that the majority of the populations from Ireland and Britain tend to be homozygous for *S* (South) haplotypes while *N* (North) haplotypes dominate in all other populations (*Figure 4—figure supplements 1* and *2*). At the Chr6 locus, all other Baltic and Atlantic populations tend to be fixed for the *N* haplotype, except a few populations from Norwegian fjords and the transition zone, which suggests that Chr6 *S* haplotype is contributing to adaptation to local ecological conditions around

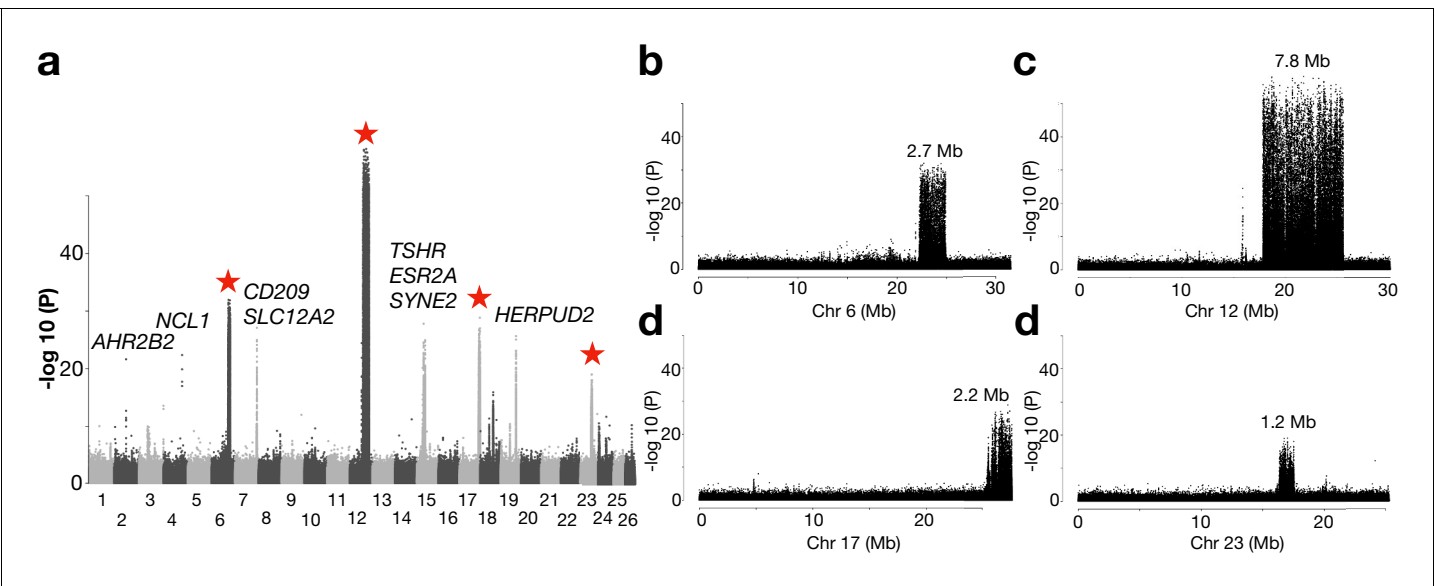

**Figure 4.** Genetic differentiation between herring populations from Ireland and Britain vs.other populations from the Northeast Atlantic. (**a**) Putative structural variants on Chr6, 12, 17 and 23 are indicated with red stars. (**b** to **e**) are zoomed-in profiles on the corresponding chromosomes. Strong genetic differentiation is observed in the following intervals: Chr6: 22.2–24.8 Mb, Chr12: 17.8–25.6 Mb, Chr17: 25.8–27.5 Mb, and Chr23: 16.3–17.5 Mb. The online version of this article includes the following figure supplement(s) for figure 4:

**Figure supplement 1.** Allele frequency of diagnostic SNPs at four putative structural variants across populations of Atlantic herring and one Pacific outgroup.

**Figure supplement 2.** Heatmap of the genotypes for diagnostic SNPs based on individual whole-genome sequencing data at four putative structural variants.

**Figure supplement 3.** Frequencies of two estimated haplotypes in each population at four putative structural variants on Chr6 (**a**), Chr12 (**b**), Chr17 (**c**), and Chr23 (**d**) across the geographical distribution.

**Figure supplement 4.** Neighbor-joining tree based on genotypes of all SNPs in each of the four putative structural variants respectively on Chr6 (**a**), Chr12 (**b**), Chr17 (**c**), and Chr23 (**d**).

**Figure supplement 5.** Comparisons of p-values and delta allele frequencies (DAF) for the contrast shown in *Figure 4*.

Ireland and Britain, and occurs at a particularly high frequency in populations spawning near the Isle of Man in the Irish Sea, the Downs near the English Channel and in the Celtic Sea (*Figure 4—figure supplement 3a*). The most striking differentiation was observed on Chr12, where the -log10(*P*) value exceeded 50 (*Figure 4c*). In fact, it coincides with a recently reported inversion, which is potentially associated with adaptation to water temperature at spawning (*Pettersson et al., 2019*) or at the larval stage over the winter months (*Fuentes-Pardo et al., 2019*). The haplotype frequencies among populations supports this hypothesis (*Figure 4—figure supplement 3b*). The individual sequence data reveal that recombinant versions of this inversion are frequent in the Baltic Sea, but the exact breakpoints of the recombined regions within the inversion are variable (*Figure 4—figure supplement 2*).

The putative inversion on Chr17 shows a similar pattern as the Chr6 polymorphism in the southernmost populations in East Atlantic where the *S* haplotype is predominant, but some populations near the transition zone also carry a large proportion of the *S* haplotype (*Figure 4—figure supplement 3c*). No population is fixed for the *S* haplotype at the Chr23 locus, although S is still the dominant haplotype in the majority of the southernmost populations (*Figure 4—figure supplement 3d*). The association between the haplotype frequencies and latitude across populations for the Chr12, Chr17, and Chr23 variants may underlie adaptation to ecological factors that are related to water temperature (see *Figure 1—figure supplement 1*). The Chr6 locus seems to be associated with ecological adaptation specifically to habitats around Ireland and Britain.

We constructed a neighbor-joining tree based on all SNPs from each of the four loci and estimated the nucleotide diversity of each haplotype. We found that, in general, both *N* and *S* haplotypes at each locus show lower nucleotide diversity in the range 0.11% to 0.31% than the genome average (~0.3%) and *S* haplotypes tend to be more variable than *N* haplotypes (*Figure 4—figure supplement 4*). This suggests that genetic recombination has been restricted between haplotypes and that *N* haplotypes are younger than the S haplotypes or have been more affected by population bottlenecks, possibly before or during the last glaciation.

## Signatures of selection in herring from the North Atlantic Ocean

Another pattern was that populations from the North Atlantic Ocean formed two major groups (*Figure 2b*; *Figure 2—figure supplement 1*). These groups include spring- and autumn-spawning populations from Nova Scotia, Newfoundland, Greenland, Iceland and those representing the Norwegian Spring-Spawning (NSS) herring (#41 and 44). The latter is the most abundant herring stock in the world with a current adult breeding stock of $>2\times10^{11}$ individuals (*ICES, 2019*). The stock spawns along the Norwegian coast in early spring, followed by a northward larval drift and later feeding migration into the Norwegian Sea. After feeding, the majority of the stock overwinters in the fjords of northern Norway (*Dragesund et al., 1997*). We designed a contrast between this group of 10 population samples and all other 21 population samples but excluded those from the Baltic Sea and Norwegian fjords. The whole-genome screen revealed, as expected, the four major loci on Chr6, 12, 17, and 23, that were significant in the partially overlapping former contrast (*Figure 5a*). However, the most striking signal of selection in the contrast involving the North Atlantic herring is a locus on Chr2 overlapping a single gene: *Aryl Hydrocarbon Receptor 2B2 (AHR2B2)* (*Figure 5a*). Four of the ten top SNPs at this locus are missense mutations, including the one (Ser346Thr) that by far showed the strongest genetic differentiation ($p<10^{-45}$; *Figure 5b*), suggesting that it is likely to be a causal variant. The non-reference allele Thr346 occurs at a high frequency in all population samples from the North Atlantic Ocean whereas the reference allele Ser346 dominates in all other population samples (*Figure 5—figure supplement 1a*). However, the Thr346 allele is widespread and there is a tendency for a North-South cline both in the Baltic Sea and around Ireland and Britain with a slightly higher frequency of Thr346 in the North.

Besides the *AHR2B2* locus, we noted several other highly significant loci not identified in other contrasts. The most significant SNP on Chr7 ($p<10^{-24}$; *Figure 5a*) is located ~1 kb upstream of *solute carrier family 12 (sodium/chloride transporter) member 2 (SLA12A2/NKCC1)*, a gene that shows altered expression in response to changes in salinity in spawning sea lamprey (*Ferreira-Martins et al., 2016*). Another interesting locus is *THRB (thyroid hormone-receptor beta)* on Chr19 encoding a medically important nuclear receptor. Mutations in *THRB* cause generalized thyroid hormone resistance in human (*Ferrara et al., 2012*) and a recent CRISPR study in zebrafish revealed its function in photoreceptor development (*Deveau et al., 2019*). At the herring *THRB* locus, all

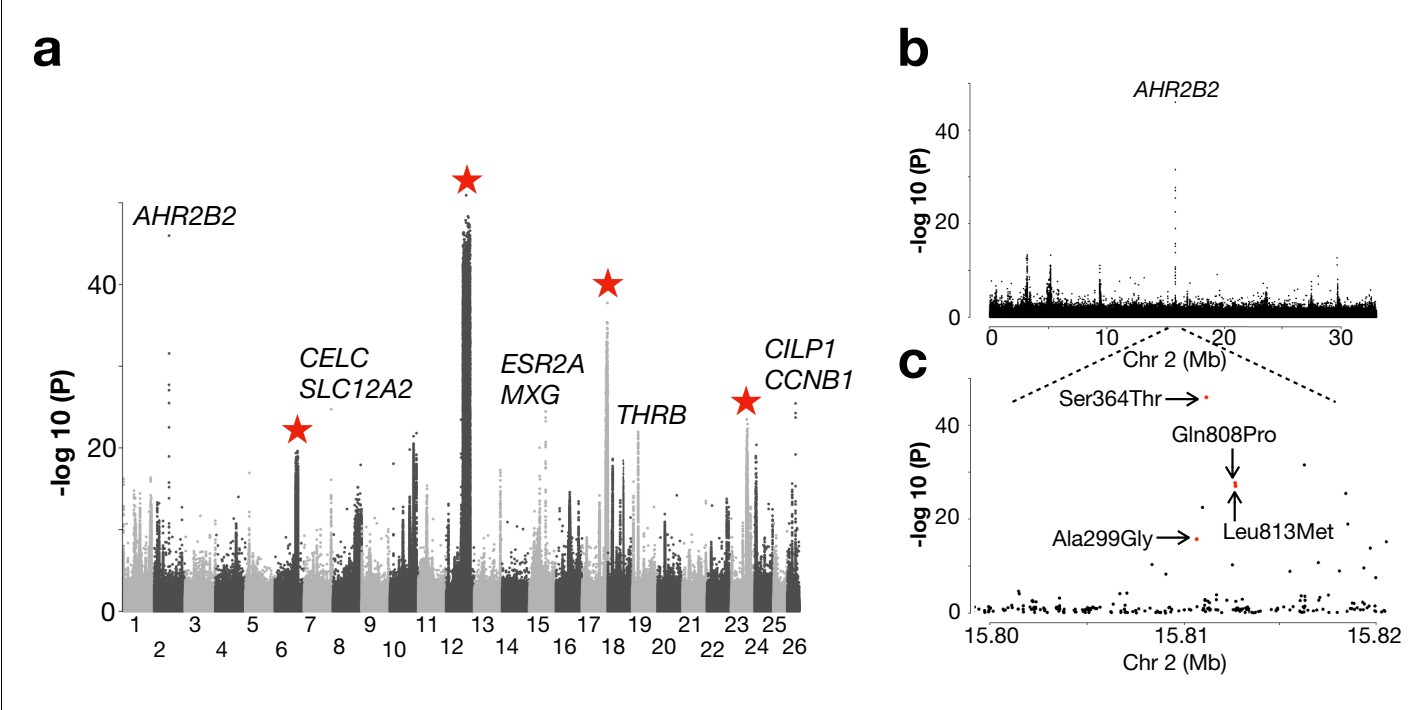

**Figure 5.** Genetic differentiation between Atlantic herring from the North Atlantic vs. waters around Ireland, Britain, and the transition zone. (a) Genome-wide screen. (b) Genetic differentiation on Chr2 and a zoom-in profile of the *AHR2B2* locus. Red dots refer to the four missense mutations. The online version of this article includes the following figure supplement(s) for figure 5:

**Figure supplement 1.** The missense mutation (Ser346Thr) in *AHR2B2* and a multi-species amino acid alignment of AHR sequences across vertebrate species.

**Figure supplement 2.** Heatmap of allele frequencies based on pooled samples for SNPs in the interval 6.35 Mb to 6.37 Mb on chromosome 19 with absolute delta allele frequencies above 0.4 in the contrast shown in *Figure 5*.

**Figure supplement 3.** Comparisons of p-values and delta allele frequencies (DAF) for the contrast shown in *Figure 5*.

populations from high salinity conditions (34–35 PSU) are fixed for an Atlantic haplotype whereas, with a few exceptions, a 'Baltic' haplotype, which must have been introgressed from the sister species Pacific herring, dominates in all populations from the Baltic Sea, the transition zone and Norwegian fjords (*Figure 5—figure supplement 2*). The only sequence difference between this introgressed haplotype and the haplotype present in Pacific herring from Vancouver is a missense mutation in *THRB*, Gln40His.

## Identification and characterization of inversions using PacBio long reads

PacBio long read sequencing of size selected fragments (>40 kb) from one male individual from the Celtic Sea was used in an attempt to validate candidate inversions on Chr6, 12, 17, and 23 (*Figure 4*). This individual was not informative for the previously documented inversion on Chr12 (*Pettersson et al., 2019*) because it was homozygous for the S haplotype present in the reference assembly. PacBio sequencing provided conclusive evidence for the presence of inversions on Chr6 and 17 (*Figure 6*). On Chr6, the distal breakpoint is at 24,868,581 bp and the proximal breakpoint lies between 22,280,036 bp and 22,290,000 bp. Its exact position is difficult to deduce due to complexity of the region and because the reference assembly is not complete at the proximal breakpoint. A representative PacBio read spanning the proximal breakpoint on Chr6 is shown in *Figure 6a* where the read starts within the inversion in the orientation toward the distal breakpoint, continues across the proximal breakpoint, and ends outside the inversion. This pattern is supported by 34 reads. *Figure 6b* illustrates the Chr17 inversion where a PacBio read maps before the proximal breakpoint and continues to the sequence proximal to the distal breakpoint in the reference assembly. There are 36 reads supporting this pattern with 25,805,445 bp as the proximal breakpoint and 27,568,511 bp as the distal breakpoint. The inserted dotplot shows the presence of a 5 kb

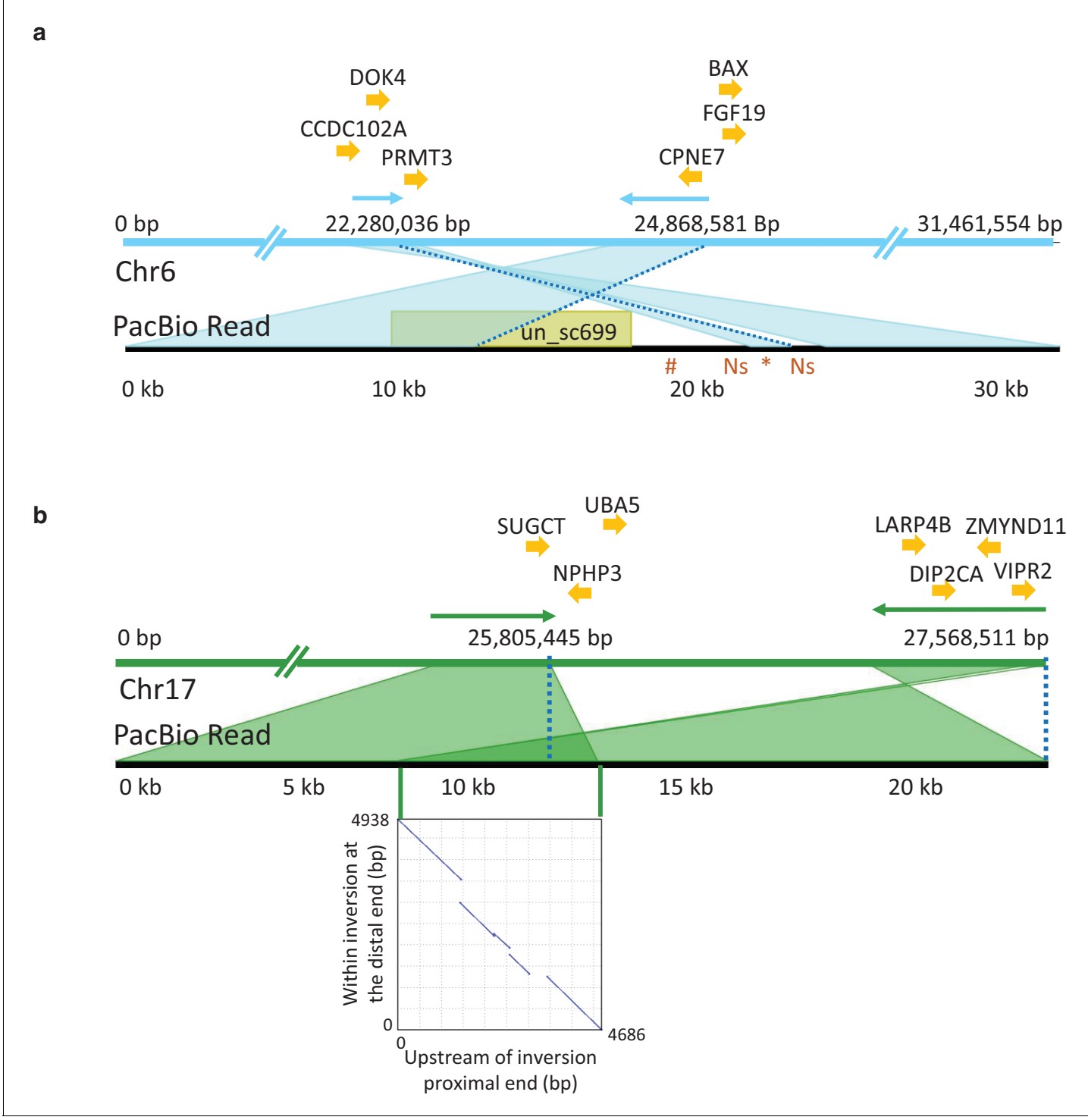

**Figure 6.** PacBio long reads defining inversion breakpoints on chromosomes 6 and 17. (**a**) Chromosome 6. The orientation of an informative read is indicated with arrows above the Chr6 reference panel. The breakpoints are indicated with dotted blue lines. The region just outside the proximal breakpoint is misassembled in the reference and results in a 1.7 kb pseudo-inversion (marked by an asterisk below the read panel) flanked by 499 bp of Ns. It also results in the finding that nearly 10 kb of the PacBio read maps to unplaced_scaffold699 (shown in yellow colored box) and nearly 5 kb (indicated with hash symbol below the read panel) that is not present in the assembly. Genes surrounding the breakpoints are indicated above the reference panel. *CCDC102A*: coiled-coil domain containing 102A, *DOK4*: docking protein 4, *PRMT3*: protein arginine methyltransferase 3, *CPNE7*: copine-7-like, *BAX*: apoptosis regulator BAX-like, *FGF19*: fibroblast growth factor 19. (**b**) Chromosome 17. The correspondence between the PacBio reads and the reference assembly is indicated with arrows above the reference panel. The breakpoints are indicated with dotted blue lines. The overlap

*Figure 6 continued on next page*

*Figure 6 continued*

of the read to both the proximal and distal breakpoints is shaded in dark green and is caused by a 5 kb duplication that is present before the proximal and distal breakpoints. The dotplot illustrating this duplication is shown in the inserted graph. Genes in the near vicinity of the breakpoints are shown above the reference sequence. *SUGCT*: succinyl –CoA: glutarate-CoA transferase, *UBA5*: ubiquitin-like modifier activating enzyme 5, *NPHP3*: nephronophthisis 3, *ZMYND11*: zinc finger, MYND-type containing 11, *LARP4B*: La ribonucleoprotein 4B homolog, *DIP2CA*: disco-interacting protein two homolog Ca, *VIPR2*: vasoactive intestinal peptide like receptor 2.

duplicated sequence in the reference N haplotype, located with an opposite orientation at the two breakpoints. This 5 kb sequence is present as a single copy in the S haplotype of the PacBio individual, implying that the N haplotype is most likely the derived state, and that the inversion and duplication occurred during the same meiotic event. This interpretation is consistent with the higher nucleotide diversity among S haplotypes (0.31% vs. 0.24%, *Figure 4—figure supplement 4c*). There is a total of 44 and 66 genes present within the inverted regions of Chr6 and 17, respectively. The inversion breakpoints do not disrupt any coding sequence, but the proximal breakpoint of the Chr6 inversion is located very close to exon 1 of the methyl transferase 3 (*PRMT3*) gene suggesting that the inversion might affect its expression pattern. It is a highly conserved gene in vertebrates and involved in bone development in human (*Min et al., 2019*). Complete lists of genes within the inverted regions and their flanking sequence are provided in *Supplementary file 6*.

The PacBio data only provided suggestive evidence for an inversion on Chr23. A proximal inversion breakpoint was indicated at 16,378,067 bp, but the characterization of the distal breakpoint was hampered because the individual used for the reference assembly (*Pettersson et al., 2019*) is most likely heterozygous for this inversion.

## Discussion

The first population genetic studies on Atlantic herring in the early 1980s were based on a dozen allozyme markers and revealed the unexpected result of no significant allele frequency differences even between Atlantic and Baltic herring considered as subspecies (*Andersson et al., 1981*; *Ryman et al., 1984*). Subsequent studies using microsatellites and small number of SNPs largely confirmed this observation but with occasional genetic markers showing genetic differentiation (*Larsson et al., 2010*; *Limborg et al., 2012*). Whole-genome sequencing has completely changed the picture, we now find striking genetic differentiation but at a limited number of loci (*Figures 3–5*). Our previous analysis (*Lamichhaney et al., 2017*) revealing a striking difference between the observed distribution of $F_{ST}$ values and the one expected under neutrality implies that essentially all loci highlighted as highly differentiated in the present study are due to adaptation. Our data are consistent with a model where the Atlantic herring has a genomic 'tool box' of a few hundred major loci associated with ecological adaptation to local climate conditions such as water temperature, salinity, light conditions, and preferred spawning time. We propose that this genetic adaptation is related to life history parameters such as migratory behavior, timing of spawning, and ecological conditions during larval development. The Atlantic herring has a high fecundity where a female can release tens of thousands of eggs at spawning. A perfect match between genotype and environmental conditions gives potential for a highly successful reproduction.

An important practical implication of the current study is that sequence variants associated with ecological adaptation best reveal the population structure of the Atlantic herring (*Figure 2b*), whereas neutral markers are less useful for population identification (*Figure 2a*). A PCA analysis based on only about 800 SNPs out of the more than 10 million SNPs detected in the study distinguish all major groups of Atlantic herring and reveal allele frequency differences between subpopulations within major groups. For instance, the Ser346Thr mutation in *AHR* alone can efficiently distinguish the North Atlantic Ocean groups from all other groups of herring (*Figure 5—figure supplement 1a*). The current study has established the diagnostic SNP markers that may be used to improve the collation of data for fisheries stock assessment. This is necessary in order to identify and address the mismatches between biological populations and management units or stocks, which can hamper the development and implementation of effective management of this important natural resource (*Cope and Punt, 2011*; *Hintzen et al., 2015*).

The genetic architecture underlying multifactorial traits and disorders has been intensively studied in recent years, in particular based on human genome-wide association studies (GWAS), and the general finding is that most traits are highly polygenic with large number of loci each explaining a tiny fraction of the genetic variance (*Goddard et al., 2016*). This is usually referred to as consistent with a quasi-infinitesimal model, because the data largely conform with the predictions of Fisher's classical infinitesimal model (*Fisher, 1919*) which constituted the foundation for quantitative genetics theory. A prime example of a trait consistent with the quasi-infinitesimal model is stature in human. A study based on ~700,000 individuals revealed more than 3000 variants affecting this trait but these did not explain more than ~25% of the variance (*Yengo et al., 2018*), implying that it must be thousands of segregating sites affecting stature in humans. The genetic architecture underlying ecological adaptation in Atlantic herring is strikingly different. It is polygenic but the number of loci showing strong genetic differentiation is in the hundreds and not in the thousands. However, it is possible that we only see the tip of the iceberg and there may be a large number of loci with tiny effects that also affect ecological adaptation. The herring data nevertheless deviates from the expected pattern under a strict infinitesimal model because this theory predicts that a response to natural selection should only cause minor changes in allele frequencies. This is because each individual locus only explains a tiny fraction of the genetic variance and should not result in major shifts in allele frequencies, such as those reported here. Allele frequencies may in fact shift substantially under the infinitesimal model but due to genetic drift, mutation or migration (*Barton et al., 2017*) which cannot explain the pattern we see here as the effect is restricted to a small portion of the genome. An important difference between the genetic architecture underlying variation in human stature and ecological adaptation in herring, is that the signatures of selection in herring reflect evolutionary processes that may have happened over many generations. For instance, the herring has adapted to the brackish Baltic Sea after it was formed about 10,000 years ago (*Andrén et al., 2011*). In contrast, the human GWAS data reflect standing genetic variation, which includes many slightly deleterious variants, the majority of these will never contribute to a selection response for a fitness trait. This interpretation is supported by the report that negative (or purifying) selection is common for genetic variants associated with human complex traits (*Zeng et al., 2018*) The genetic architecture underlying ecological adaptation in herring is in fact similar to the genetic architecture underlying skin color in humans, a trait affected by strong natural selection, for which four major loci were estimated to explain 35% of the total variance in an admixed population (*Beleza et al., 2013*).

The genetic architecture underlying ecological adaptation detected here makes perfect sense in light of the ecological challenges involved. The most dramatic differences in biotic and abiotic factors occur between the marine Atlantic Ocean and the brackish Baltic Sea which differ in regard to salinity, variation in water temperature over the year (*Figure 1—figure supplement 1*), light conditions (*Hill et al., 2019*), plankton production and presence of predators. In this contrast, we document hundreds of loci showing significant genetic differentiation. In other contrasts between geographically distinct ecoregions, the numbers of detected loci are much fewer, in the range 10–30 (*Figures 3–5*).

It has been exceedingly challenging to move from GWAS peaks to the identification of causal variants in human and other species, despite enormous efforts. The reason is that strong linkage disequilibrium among multiple sequence variants showing an equally strong association with phenotype makes it difficult or impossible to provide genetic evidence for causality. In contrast, the genome-wide screens in Atlantic herring immediately reveal strong suggestive evidence of causality at some loci. A striking example concerns *rhodopsin* (*RHO*), showing strong genetic differentiation between Atlantic and Baltic herring (*Figure 3a*). A missense mutation in this gene (Phe261Tyr) has a critical role for adaptation to the red-shifted light conditions in the Baltic Sea (*Hill et al., 2019*). Interestingly, about one third of all fish adapted to red-shifted light conditions in freshwater and brackish water carry Tyr261 like the Baltic herring, an amazing example of convergent evolution (*Hill et al., 2019*). The present study revealed at least two additional examples of outstanding candidates for being causal mutations, the non-coding single-base change upstream of *SOX11B* (*Figure 3—figure supplement 2b*) and the Ser346Thr missense mutation at *AHR2B2* (*Figure 5b*). These are candidate causal mutations based on far stronger statistical support than any other sequence variant in the corresponding regions. How can genomic regions showing signals of selection in these cases be so narrow? There are three possible explanations, either it is because (i) there is an extremely high rate of recombination in the region, (ii) these are 'old' polymorphisms that have segregated for such a long

time that linkage disequilibrium to flanking markers has eroded leaving the causal variant in splendid isolation, or (iii) the same causal variant has occurred by mutation multiple times on different haplotypes. We can exclude option (i) because neither meiotic recombination or LDhat data indicate an exceptionally high recombination rate in this region (*Pettersson et al., 2019*). Although the mutation rate in the Atlantic herring is exceptionally low (*Feng et al., 2017*), in each generation every nucleotide site in the herring genome is hit by many mutations due to the enormous population size. Thus, herring is not a species where evolution is restricted by lack of genetic variation and therefore has to rely on standing genetic variation. The observation of sequence variants with outstanding strength of associations, implying substantial functional effects, is a deviation from the infinitesimal model.

A unique observation in this study is the Ser346Thr missense mutation in the *aryl hydrocarbon receptor 2B2* (*AHR2B2*) gene. Thr346 must be the derived allele since all tested Pacific herring were homozygous Ser346. Furthermore, residue 346 is located in the highly conserved PAS-B domain and Ser346 is highly conserved among vertebrates (*Figure 5—figure supplement 1b*), suggesting a functional significance of Ser346Thr. AHR2B2 belongs to the basic helix-loop-PER-ARNT-SIM (bHLH-PAS) family of transcription factors. AHR is cytosolic signal sensor that binds planar aromatic hydrocarbons and is then translocated to the nucleus where it forms a heterodimer with the AHR nuclear translocator (ARNT) and affect transcription (*Schulte et al., 2017*; *Tischkau, 2020*). The endogenous ligands of AHR include tryptophan-derived metabolites and dietary indoles, while the most well-known exogenous high-affinity ligand is dioxin (*Schulte et al., 2017*). In human medicine, AHR function is of relevance for toxicity response, inflammation, autoimmune disorders, cancer, circadian clock and for sensing bacterial virulence and thereby activate antibacterial responses and control sepsis (*Schulte et al., 2017*; *Tischkau, 2020*). It has recently been reported that a population of killifish has been able to colonize highly polluted waters due to the presence of a deletion knocking out AHR function (*Oziolor et al., 2019*). In herring, the derived Thr346 allele occurs at a high frequency in the least polluted waters in the North Atlantic Ocean, for example around Greenland and Iceland, suggesting that it is unlikely that this adaptation is related to the presence of exogenous metabolites. A serine to threonine change is a conservative amino acid substitution since they are both polar, neutral amino acids with a hydroxyl or methyl side chain, respectively. This suggests that this missense mutation fine-tunes rather than disrupts AHR function. Billions of herring in the North Atlantic Ocean are homozygous for Thr346 (*Figure 5—figure supplement 1b*). Genetic adaptation in a poikilotherm organism exposed to a wide range of water temperatures (*Figure 1—figure supplement 1*) must involve mutations affecting protein function in a temperature-dependent manner. Interestingly, residue 346 occurs in the ligand-binding PAS-B domain and Ser346Thr may contribute to cold adaptation by affecting ligand-binding as it dominates in populations exposed to the coldest waters (*Figure 5—figure supplement 1a*).

The importance of inversions underlying phenotypic variation and ecological adaptation is a hot topic in evolutionary genomics (*Wellenreuther and Bernatchez, 2018*). Inversions suppress recombination facilitating the evolution of 'supergenes' composed of a cluster of co-adapted alleles. Characterization of inversions using short read sequence data is challenging because inversion breakpoints are often embedded in repeat regions. Here we have successfully used PacBio long reads and generated conclusive evidence for inversions on Chr6 and 17 in herring (*Figure 6*), and suggestive evidence for a third one on Chr23. Furthermore, the individual used for PacBio sequencing was homozygous for the reference allele at the inversion on Chr12 documented in our recent study (*Pettersson et al., 2019*). The characteristic features of these four inversion polymorphisms are that (i) the inversion breakpoints do not disrupt coding sequence (conclusive evidence only for Chr6 and 17 so far), (ii) no allele is lethal in the homozygous condition, (iii) the inversions suppress recombination but not completely, as we find evidence for recombinant haplotypes (*Figure 4—figure supplement 2*), and (iv) none of the inversions is of recent origin as indicated by substantial nucleotide diversity within each haplotype group, in the range 0.11–0.31% (*Figure 4—figure supplement 4*). The strong genetic differentiation among populations for each of the four inversions (*Figure 4—figure supplement 3*) implies that these are supergenes important for environmental adaptation. It is tempting to speculate that several of these, if not all, are directly or indirectly related to the water temperature at spawning due to the relatively strong correlation between water temperatures and haplotype frequencies (*Figure 1—figure supplement 1* and *Figure 4—figure supplement 3*). The higher nucleotide diversity for the Southern haplotype groups at each locus

suggests that these may represent the ancestral state and that the derived alleles provided a selective advantage during an expansion of Atlantic herring into colder waters.

In conclusion, ecological adaptation in the Atlantic herring is associated with large shifts in allele frequencies at loci under selection. The underlying cause is both (i) sequence variants of major importance, like Phe261Tyr in *rhodopsin* (*Hill et al., 2019*), and (ii) haplotypes of major importance carrying multiple sequence changes. The most extreme form of the latter is megabase inversions/supergenes such as those reported here (*Figures 4* and *6*).

## Materials and methods

### Sample collection and genome resequencing

The study is based on sequence data generated in this study as well as those reported in our previous studies (*Supplementary files 1* and *2*). Tissue from 35 to 100 fish were collected from different localities in the Baltic Sea, Skagerrak, Kattegat, North Sea and the waters surrounding Ireland and Britain, the Atlantic Ocean, and the Pacific Ocean. Genomic DNA was prepared using standard methods, pooled and individual sequencing was carried out as previously described (*Martinez Barrio et al., 2016*). Illumina 2 × 150 bp short read sequences for the new populations in this study were analysed together with read sets that were generated for populations used previously.

### Alignment and variant calling

Paired-end reads including the data from previous studies (*Lamichhaney et al., 2017*; *Martinez Barrio et al., 2016*) were aligned to the reference genome (*Pettersson et al., 2019*) using BWA v0.7.17 component BWA-MEM (*Li, 2013*). Postprocessing of alignment was handled with Samtools v1.6 (*Li et al., 2009*). Merging libraries of the same sample and marking duplicates were done using Picard toolkit v2.10.3 (*Picard toolkit, 2019*). Raw genomic variants were called using GATK UnifiedGenotyper for pooled sequencing data and GATK HaplotypeCaller for individual sequencing data (*McKenna et al., 2010*). Stringent filtration of the raw variants was done following the criteria "QD <5.0 || FS >40.0 || MQ <50.0 || MQRankSum <−4.0 || ReadPosRankSum < −3.0 || ReadPosRankSum > 3.0 || DP <100 || DP >5400' for pooled data and "QUAL < 100 || MQ <50.0 || MQRankSum <−4.0 || ReadPosRankSum < −4.0 || ReadPosRankSum > 4.0 || QD <5.0 || FS >40.0 || DP <100 || DP >5500' for individual data using GATK VariantFiltration and SelectVariants. All the cut-offs were determined according to the genome-wide distribution of each parameter. To explore haplotypes in the downstream studies, we phased the individual data using BEAGLE v4.0 (*Browning and Browning, 2007*).

### Analysis of population divergence

The allele frequency of pool-seq data are calculated based on the counts of reference and alternate reads at each polymorphic site. However, using raw read count could introduce bias in particular population or genomic regions, due to varied sample size across the pools or varied coverage along the genome. To minimize the technical bias, we carried out the $N_{eff}$ allele count correction (*Bergland et al., 2014*; *Fuentes-Pardo et al., 2019*) before the calculation of allele frequency, following the formula: $N_{eff} = (2 n * RD- 1) / (2 n + RD)$, where RD is raw read depth and n is the number of individuals in a given pool.

Due to baseline differences in the contrasts, the $\chi^2$ p-values are somewhat inflated (λ in the range 1.55–1.92). However, as can be seen in *Figure 3—figure supplement 4*, *Figure 4—figure supplement 5* and *Figure 5—figure supplement 3*, the signals extend far above the level of inflation, the p-values are strongly correlated with delta allele frequency and the significance threshold delineate markers that are clearly distinct from the background for all contrasts. In the text, in main figures and in *Supplementary files 4* and *5*, we present λ-adjusted p-values, while Figure supplements show raw values.

To examine population structure and genetic differentiation among the pools, we performed principal component analysis (PCA) based on corrected allele frequencies of two sets of markers showing (i) no significant genetic differentiation and (ii) highly significant genetic differentiation. The two sets of markers were established based on the genome-wide distribution of the standard

deviation (std) of allele frequencies across populations. Based on the genome-wide distribution, we defined SNPs with $0.02 \leq$ std $\leq 0.08$ and std $\geq 0.2$ as undifferentiated markers and highly differentiated markers, respectively (*Figure 2—figure supplement 3*). Minor allele frequency was required to be >0.01 in order to exclude monomorphic sites and rare variants. We further down-sampled the datasets by retaining only one marker within every 1 kb of the genome for low differentiated markers and 10 kb for highly differentiated markers, in order to weaken the weight from genetically linked regions; the difference for the two sets are justified because linkage disequilibrium is decaying rapidly in the herring genome except in some of the regions under selection (*Martinez Barrio et al., 2016*). In the final PCA, we used 169,394 SNPs as undifferentiated markers and 794 as highly differentiated markers. Sample 38 was excluded from the PCA analysis due to a potential sequencing quality issue. The location and pool-wise frequency estimates for the 794 markers can be found in *Supplementary file 7*.

Additionally, we constructed a genetic distance tree based on ~3.2 million whole-genome SNPs after filtering out missing data, using PHYLIP package v3.696 (*Felsenstein, 1995*). We randomly sampled 1000 replicate datasets using Seqboot bootstrap method and computed pairwise Nei distances using Genedist. For each replicate, a neighbor-joining tree was inferred using Neighbor. A final consensus tree was produced from all trees with Consensus using the Majority Rule method. Tree visualization was done in FigTree v1.4.4 (*Rambaut, 2014*).

Pairwise $F_{ST}$ in 5 kb sliding windows was calculated using samtools (*Li et al., 2009*) and PoPoolation2 (*Kofler et al., 2011*) with the following pipeline and parameters: samtools mpileup -B -q 20 -b; java -ea -Xmx40g -jar mpileup2sync.jar `–fastq-type` sanger `–min-qual` 20 `–threads` 5; perl fst-sliding.pl `–min-count` 10 `–min-coverage` 10 –max-coverage 500 `–min-covered-fraction` 0.7 `–window-size` 5000 `–step-size` 5000. The sample size of each pool was provided to correct the sequencing depth.

## Identifying genetic signature of selection

To maximize the statistical power and minimize the within-population variance, we grouped populations into superpools according to their environmental variables or phenotype, and made contrasts between superpools (*Supplementary file 3*). For each contrast, we summed up the read counts separately for reference and alternate alleles in each superpool, and performed $2 \times 2$ contingency $\chi^2$ test between superpools for every SNP ($\sim 6 \times 10^6$ per contrast). Bonferroni correction of the p values was calculated in each contrast, and adjusted $p < 1 \times 10^{-10}$ was determined as the significance threshold. We clustered the signals as independent loci as described (*Martinez Barrio et al., 2016*), by requiring that such regions (1) should be at least 100 bp in length, (2) have at least two SNPs in each contrast reaching a significance threshold of raw $p < 1 \times 10^{-20}$ in the $\chi^2$ test, and (3) have a minimal distance of 100 kb from the neighboring loci. We summarized the independent loci in each contrast, and intersected individual SNPs, in order to identify replicated signals between the two contrasts of adaptation to salinity and spawning season, respectively.

## Population genetic analysis of putative structural variants

Diagnostic markers at the four putative structural variants on Chr6, 12, 17, and 23 were selected for investigating haplotype frequencies across populations in pooled and individual sequencing data. Independent neighbor-joining trees were constructed based on all SNPs from each of the putative structural variants using PLINK v1.90b4.9 (*Purcell et al., 2007*) and neighbor from the PHYLIP package v3.696 (*Felsenstein, 1995*). We estimated nucleotide diversity of each haplotype at the SVs using vcftools v0.1.16 (*Danecek et al., 2011*).

## PacBio long read sequencing for characterization of inversion breakpoints

A male Atlantic herring was captured in November 25, 2019 in the Celtic Sea (N51.6, W6.5) and flash frozen in liquid nitrogen. High molecular weight DNA was extracted from testis using the Circulomics Nanobind Tissue Big DNA Kit (NB-900-701-001). Size selection for molecules over 40 kb using BluePippin (SageScience), and libraries for PacBio sequencing was made using the manufacturer's protocol. Sequencing was carried out on one PacBio Sequel II 8M SMRT Cell for 15 hr to generate 125 Gb CLR sequence data. Subreads were aligned to the herring reference assembly

(*Pettersson et al., 2019*) using Minimap2 aligner (*Li, 2018*). Only reads with mapping quality greater than 20 were used for further analysis. Average genome coverage was measured using Samtools (*Li et al., 2009*) and was found to be 110x. Structural variants were called with Sniffles (*Sedlazeck et al., 2018*). The alignment was viewed using IGV (*Robinson et al., 2011*) and Ribbon (*Nattestad et al., 2016*) for the inversion signal. Mummer was used to build the dotplot (*Delcher et al., 2002*).

## Acknowledgements

We are grateful to Erik Enbody for insightful comments on the manuscript and the WESTHER project, the Irish Marine Institute, Marine Scotland Science, the Northern Ireland Agri-Food and Biosciences Institute, H Ojaveer and Thomas Gröhsler (Thünen Institute of Baltic Sea Fisheries, Rostock, Germany) for sample collections. The National Genomics Infrastructure (NGI)/Uppsala Genome Center and UPPMAX provided service in massive parallel sequencing and computational infrastructure. Work performed at NGI/Uppsala Genome Center has been funded by RFI/VR and Science for Life Laboratory, Sweden.

## Additional information

### Funding

| Funder | Grant reference number | Author |
| --- | --- | --- |
| Knut och Alice Wallenbergs Stiftelse | KAW scholar | Leif Andersson |
| Vetenskapsrådet | Senior professor | Leif Andersson |
| Research Council of Norway | GENSINC, project 254774 | Florian Berg Arild Folkvord Leif Andersson |
| Danish Ministry of Food, Agriculture and Fisheries | 33113-B-19-154 | Dorte Bekkevold |
| European Fisheries Fund | 33113-B-19-154 | Dorte Bekkevold |

The funders had no role in study design, data collection and interpretation, or the decision to submit the work for publication.

### Author contributions

Fan Han, Formal analysis, Investigation, Visualization, Methodology, Writing - original draft, Writing - review and editing; Minal Jamsandekar, Formal analysis, Investigation, Visualization, Writing - review and editing; Mats E Pettersson, Data curation, Formal analysis, Investigation, Visualization, Methodology, Writing - review and editing; Leyi Su, Angela P Fuentes-Pardo, Investigation, Visualization, Writing - review and editing; Brian W Davis, Supervision, Investigation, Writing - review and editing; Dorte Bekkevold, Conceptualization, Resources, Writing - review and editing; Florian Berg, Michele Casini, Geir Dahle, Edward D Farrell, Resources, Writing - review and editing; Arild Folkvord, Conceptualization, Resources, Supervision, Funding acquisition, Project administration, Writing - review and editing; Leif Andersson, Conceptualization, Resources, Supervision, Funding acquisition, Validation, Investigation, Visualization, Methodology, Project administration, Writing - review and editing

### Author ORCIDs

Mats E Pettersson (iD) http://orcid.org/0000-0002-7372-9076
Florian Berg (iD) http://orcid.org/0000-0003-1543-8112
Arild Folkvord (iD) http://orcid.org/0000-0002-4763-0590
Leif Andersson (iD) https://orcid.org/0000-0002-4085-6968

### Decision letter and Author response

Decision letter https://doi.org/10.7554/eLife.61076.sa1

Author response https://doi.org/10.7554/eLife.61076.sa2

## Additional files

### Supplementary files
- Supplementary file 1. Specimen information of 53 pooled samples.
- Supplementary file 2. Specimen information of 55 individual sequencing samples.
- Supplementary file 3. Definition of how superpools were formed by pooling population samples of major groups of herring. Sample names correspond to those given in *Supplementary file 1*.
- Supplementary file 4. List of SNPs showing strong genetic differentiation between superpools of Atlantic and Baltic herring. The criteria for inclusion was raw $p<10^{-30}$ for both comparisons (among spring-spawners and among autumn-spawners) or raw $p<10^{-50}$ for a single comparison. These are the SNPs colored in red or black in *Figure 3—figure supplement 1*. Raw p-values are given in columns E and G and corrected p-values in columns F and H.
- Supplementary file 5. List of loci showing strong genetic differentiation between superpools of spring and autumn-spawning herring. The criteria for inclusion was raw $p<10^{-30}$ for both comparisons (among Atlantic populations and among Baltic populations) or raw $p<10^{-50}$ for a single comparison. These are the SNPs colored in red or black in *Figure 3—figure supplement 3*. Raw p-values are given in columns E and G and corrected p-values in columns F and H.
- Supplementary file 6. Genes within inverted regions and 200 kb flanking inversion breakpoints on chromosome 6 and 17 in Atlantic herring.
- Supplementary file 7. Reference allele frequencies for the 794 markers used in the PCA shown in *Figure 2b*.
- Transparent reporting form

### Data availability
The sequence data generated in this study is available in Bioproject PRJNA642736. Allele frequencies for each SNP in each population is available at Dryad (https://doi.org/10.5061/dryad.pnvx0k6kr). The analyses of data have been carried out with publicly available software and all are cited in the Methods section. Custom scripts used are available in Github (https://github.com/Fan-Han/Population-analysis-with-pooled-data); copy archived at https://archive.softwareheritage.org/swh:1:rev:64968a860b348f315134f396c14ffb0830c61eaa/.

The following datasets were generated:

| Author(s) | Year | Dataset title | Dataset URL | Database and Identifier |
|---|---|---|---|---|
| Andersson L | 2020 | Re-sequencing of Atlantic Herring populations and individuals | https://www.ncbi.nlm.nih.gov/bioproject/PRJNA642736/ | NCBI BioProject, PRJNA642736 |
| Pettersson M, Andersson L | 2020 | Reference allele frequencies for populations pools of Atlantic Herring (Clupea harengus) | https://doi.org/10.5061/dryad.pnvx0k6kr | Dryad Digital Repository, 10.5061/dryad.pnvx0k6kr |

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
