## [Decision Letter]

Thank you for submitting your article "Genetic architecture underlying ecological adaptation in Atlantic herring is not consistent with the infinitesimal model" for consideration by *eLife*. Your article has been reviewed by two peer reviewers, and the evaluation has been overseen by a Reviewing Editor and Patricia Wittkopp as the Senior Editor. The reviewers have opted to remain anonymous.

The reviewers have discussed the reviews with one another and the Reviewing Editor has drafted this decision to help you prepare a revised submission.

Title: We suggest changing the title to avoid raising expectations about dismissing an infinitesimal model. For example: The genetic architecture underlying ecological adaptation in Atlantic herring.

Summary:

This paper reports population sampling and a scan for selection in Atlantic herring. The importance of the work derives from the application of genetic results for assessing herring stock, and hence help safeguard sustainable fishing, from better understanding of the genetic changes associated with different marine environments (herring have adapted to brackish sea water) and with the herring's photoperiodic regulation of reproduction. The paper reports analysis of whole genome sequence data from 53 population samples, selected from the entire species distribution, and identifies multiple loci underlying ecological adaptation to different geographic areas and spawning conditions.

1) Calibrating the test statistics. The P-values for selection are not well-calibrated. The null hypothesis is that the allele frequencies are identical between subgroups, which is inappropriate because even under neutrality they will vary because of drift and shared ancestry between populations. The selection test needs to be corrected for this population structure. There are a number of approaches that could be used, e.g. post-hoc (e.g. genomic control), or model-based (e.g. a mixed model or including PCs as covariates).

2) The paper does not make a convincing case that the findings are "in conflict with the infinitesimal model for complex traits". Overall the arguments presented are over-simplistic and fail to take into account the following:

i) An important question about adaption is the distribution of selection coefficients across mutations, for example whether adaptation is driven by new beneficial mutations of large effects that are selected to fixation or if adaptation is polygenic (e.g. Prichard et al., 2010, Current Biology). If the latter, then does it matter how “poly” polygenic is? In any case, even though the authors report a number of loci with large allele frequency shifts, can they actually reject an adaptation model with some loci with large effect sizes and many others with tiny effect sizes?

ii) The "pure" infinitesimal model is something of a straw man, since we know that there are a finite number of causal variants, with substantial variation in effect size and that we expect the change in allele frequency of any variant to depend on effect size. Instead, the results from the herring analyses seem consistent with the observations made in many systems (e.g body size in dogs or skin pigmentation in humans) that strong selection can lead to rapid differentiation driven by large-effect variants, even if the within-population architecture remains relatively infinitesimal.

iii) There is a distinction between the within-population variance (which may still be infinitesimal), and the between-population variation, which may be driven by a small number of large-effect and highly differentiated variants. Imagine you had two populations with exactly the same infinitesimal genetic architecture, but that differed by a single fixed variant with large effect. Would you say that was "in conflict" with the infinitesimal model?

3) It's possible that most, if not all, of the observed differentiation is due to adaptation. If Ne is very large, then natural selection and adaptation are efficient in eliminating deleterious alleles and fixing beneficial alleles. Do the authors have a model that is consistent with the observed presumed “neutral” variation? For example, the pairwise Fst values (0.01 to 0.06) are not small – are they consistent with the “long-term effective population size” that the authors hypothesise to be determined by periods of glaciation? Also, if short-term Ne is huge then why are the alleles under selection not swept to fixation? For a very large number of breeding individuals, Ne may be more limited by linkage than by demographic factors, a phenomenon that has been termed “genetic draft” (Walsh and Lynch, 2018, page 76).

4) The authors' interpretation of the PCA results is open to question. The authors first estimate frequency differentiation on all polymorphisms, then select the least and most differentiated loci and perform PCA plots on the ascertained sets of loci (Figure 2). The authors contrast the variance explained by the first PCs and the resulting structure from the two sets of loci and claim that the set of highly differentiated loci reflect ecological niche adaptation, whereas the less differentiated loci reflect location. While it's reasonable to identify the highly differentiated variants as Ancestry Informative Markers, the comparison of the variance explained likely does not mean that much. In any dataset, if you select highly differentiated variables, you will explain more of the variance with the first PC just by construction. For almost any genomic dataset, you would be able to select a small number of SNPs such that you could cluster populations based on PCA of those SNPs, but it does not necessarily follow that those highly differentiated SNPs are actually under selection – that needs to be shown. A related question is whether the extra clusters that appear with the differentiated markers are actually visible in the higher PC of the undifferentiated markers. It would be helpful to show maybe PC 3,4 etc… for those data.

5) The block-like structures in the selection scan are interpreted as inversions or structural rearrangements that are under selection. Inversions suppress recombination in the inversion, but not in the flanking regions, so if the inversion were under selection, you expect to see the normal decay of signal around the fringes of the inversion. Or is it just that the scale is different so you can't see the decay on the plots? Is it possible that the high differentiation in these regions is due to mapping errors in the rearranged regions? For example, a duplication would lead to apparently highly differentiated allele frequencies.

6) A related concern to the first point is whether there are technical batch effects due to i) combination of data sequenced in different experiments (i.e. previously published) and ii) combination of pooled and individual sequencing data. It would be helpful to make sure that these do not further inflate the test statistics.

7) Pooling errors. The authors use pools for sequencing and use a Neff allele count correction for read depth to minimise technical bias. They then use the estimated allele count in 2x2 tables to contrast allele frequencies between (super) pools, using an arbitrary p-value threshold of 1e-10. Not observing the actual allele count in the sample can lead to an increased variance and therefore a biased test for association.

Revisions expected in follow-up work:

1) P-value calibration. The selection scan uses an inappropriate null distribution and therefore all the P-values are poorly calibrated and thus uninterpretable. Whichever approach is chosen to calibrate P-values, please assess its performance and include QQ plots in the paper. Without this it's impossible to know how many of the selection hits are actually significant.

2) Either remove the claim that the results are in conflict with the infinitesimal model ( our recommendation) or provide a more convincing case to support the claim.

3) Please explain the extent to which the observed differentiation could be due to adaptation, dealing with the points we raised in point 3.

4) Please respond to our criticism of your interpretation of the PCA results.

5) Please defend your interpretation of the block-like structures in the selection scan, in response to our point 5.

6) Include batch in the model and check that none of the PCs are predicting batch or data type. Make sure that these do not further inflate the test statistics.

7) Provide data on how pooling errors might have affected your results. Please show Q-Q plots for their statistical test and estimate the sensitivity of your conclusions with respect to the chosen P-value threshold.

---

## [Author Response]

Title: We suggest changing the title to avoid raising expectations about dismissing an infinitesimal model. For example: The genetic architecture underlying ecological adaptation in Atlantic herring

We have changed the title to Ecological adaptation in Atlantic herring is associated with large shifts in allele frequencies at hundreds of loci.

Revisions for this paper:1) Calibrating the test statistics. The P-values for selection are not well-calibrated. The null hypothesis is that the allele frequencies are identical between subgroups, which is inappropriate because even under neutrality they will vary because of drift and shared ancestry between populations. The selection test needs to be corrected for this population structure. There are a number of approaches that could be used, e.g. post-hoc (e.g. genomic control), or model-based (e.g. a mixed model or including PCs as covariates).

We have now included QQ plots, p-value vs δ allele frequency (DAF) plots and histograms of DAF for markers below and above the p-value theeshold for the various contrasts (Figure 3—figure supplement 4, Figure 4—figure supplement 5, and Figure 5—figure supplement 3). In our view, these clearly illustrate that:

i) While there is some inflation, it is marginal compared with the scale of the signals.

ii) The p-values, in spite of the inflation, are a useful measure of increasing divergence.

iii) The markers above the 10^-10^ p-value cutoff display a DAF distribution that is clearly distinct from the remaining marker set, indicating that it is a serviceable divider.

We also believe that the numerus illustrations of pool-wise frequencies (e.g. Figure 4—figure supplement 1 and 3, Figure 5—figure supplement 2) provide an intuitive translation of the p-values into absolute divergence metrics.

2) The paper does not make a convincing case that the findings are "in conflict with the infinitesimal model for complex traits". Overall the arguments presented are over-simplistic and fail to take into account the following:i) An important question about adaption is the distribution of selection coefficients across mutations, for example whether adaptation is driven by new beneficial mutations of large effects that are selected to fixation or if adaptation is polygenic (e.g. Prichard et al., 2010, Current Biology). If the latter, then does it matter how “poly” polygenic is? In any case, even though the authors report a number of loci with large allele frequency shifts, can they actually reject an adaptation model with some loci with large effect sizes and many others with tiny effect sizes?

In fact, our interpretation is that the genetic architecture is composed of hundreds of loci with major effects and many others with minor effects. This is what we wrote in the submitted version: “…there may be a large number of loci with tiny effects that also affect ecological adaptation”. But there should not be any loci with major effects under a “pure” infinitesimal model.

ii) The "pure" infinitesimal model is something of a straw man, since we know that there are a finite number of causal variants, with substantial variation in effect size and that we expect the change in allele frequency of any variant to depend on effect size. Instead, the results from the herring analyses seem consistent with the observations made in many systems (e.g body size in dogs or skin pigmentation in humans) that strong selection can lead to rapid differentiation driven by large-effect variants, even if the within-population architecture remains relatively infinitesimal.

We seem to agree that a pure infinitesimal model is not consistent with the genetic architecture for instance for body size in dogs or ecological adaptation in herring. We have now added citation to one paper supporting the view that many variants affecting complex trait in humans are affected by negative selection. Such variants contribute to standing genetic variation for complex traits but is unlikely to contribute to ecological adaptation. We also cite a paper on human skin pigmentation, a trait under selection and for which the underlying genetic architecture is similar to one for ecological adaptation in herring.

iii) There is a distinction between the within-population variance (which may still be infinitesimal), and the between-population variation, which may be driven by a small number of large-effect and highly differentiated variants. Imagine you had two populations with exactly the same infinitesimal genetic architecture, but that differed by a single fixed variant with large effect. Would you say that was "in conflict" with the infinitesimal model?

In conclusion, we think we are essentially in agreement with the reviewers and editors concerning the interpretation of the genetic architecture underlying standing variation for complex traits versus ecological adaptation due to an evolutionary process. We still think it is important to highlight the difference between the two since you often come across papers where the authors assume that the genetic architecture underlying adaptation is expected to be highly complex with reference to human GWAS data. Furthermore, if there is one species where one could expect a good agreement with an infinitesimal model also for ecological adaptation, it should be the Atlantic herring. The census population size the last 10,000 years after the last glaciation has probably been of the order of 10e12, as it is now, which means that each site in the genome is mutated thousands of times each generation. Furthermore, if there are thousands of genes affecting body size in humans and body size is only one of many traits that show phenotypic differences between Atlantic and Baltic herring there was a possibility that genetic differentiation between these two populations would appear as small shifts in allele frequencies at a huge number of sites where causal alleles forms extreme haplotype diversity, but that is not at all the case and we think it is worthwhile to highlight this finding but we acknowledge the criticism and has softened the language.

3) It's possible that most, if not all, of the observed differentiation is due to adaptation. If Ne is very large, then natural selection and adaptation are efficient in eliminating deleterious alleles and fixing beneficial alleles. Do the authors have a model that is consistent with the observed presumed “neutral” variation? For example, the pairwise Fst values (0.01 to 0.06) are not small – are they consistent with the “long-term effective population size” that the authors hypothesise to be determined by periods of glaciation? Also, if short-term Ne is huge then why are the alleles under selection not swept to fixation? For a very large number of breeding individuals, Ne may be more limited by linkage than by demographic factors, a phenomenon that has been termed “genetic draft” (Walsh and Lynch, 2018, page 76).

We agree that genetic draft likely restricts nucleotide diversity in the Atlantic herring, in addition to the effects of demography. We have added a sentence pointing out that with reference to Bruce and Walsh. We have addressed the question whether most, if not all, differentiation is caused by adaptation in a previous study where we were compare the observed distribution of Fst values with the one expected under neutrality established by simulation (Lamichanney et al., 2017). The conclusion is that essentially all loci showing strong genetic differentiation are due to adaptation. There is certainly a grey zone of loci showing modest genetic differentiation whether this is due to drift or selection, but we feel confident that the 100+ loci highlighted in the present study as highly differentiated are under selection. We have cited this previous study in the first paragraph of the Discussion.

4) The authors' interpretation of the PCA results is open to question. The authors first estimate frequency differentiation on all polymorphisms, then select the least and most differentiated loci and perform PCA plots on the ascertained sets of loci (Figure 2). The authors contrast the variance explained by the first PCs and the resulting structure from the two sets of loci and claim that the set of highly differentiated loci reflect ecological niche adaptation, whereas the less differentiated loci reflect location. While it's reasonable to identify the highly differentiated variants as Ancestry Informative Markers, the comparison of the variance explained likely does not mean that much. In any dataset, if you select highly differentiated variables, you will explain more of the variance with the first PC just by construction. For almost any genomic dataset, you would be able to select a small number of SNPs such that you could cluster populations based on PCA of those SNPs, but it does not necessarily follow that those highly differentiated SNPs are actually under selection – that needs to be shown. A related question is whether the extra clusters that appear with the differentiated markers are actually visible in the higher PC of the undifferentiated markers. It would be helpful to show maybe PC 3,4 etc… for those data.

The reviewers are absolutely right that the PCA analysis per se cannot be used as an argument that the loci showing strong genetic differentiation is under selection and that was not our intention in this section. Our writing is based on our previous analysis showing that the distribution of Fst values in Atlantic herring are strikingly different from the one expected under neutrality (cited in the Introduction, Lamichanney et al., 2017). However, to avoid any misunderstanding of this we have modified the title of this subsection to “Detection of population structure” and ends with the conclusion that a modest number of strongly differentiated markers provides a much better resolution of population structure than a large number of neutral genetic markers. This is in fact an important conclusion for future monitoring of stock development. In fact, the tight cluster of populations named as Atlantic Ocean in Figure 2A contains the most numerous stocks of Atlantic herring (about 90% of the world population) and these are separated into three distinct clusters when we use strongly genetically differentiated markers: spring-spawning herring from Atlantic Ocean, autumn-spawning populations from the Atlantic Ocean and populations from the water around Ireland and Britain. We highlight this observation in the revised text.

PC:s 3 and 4 do not resolve any further groupings, in our view they mostly represent technical noise (see Author response image 1; sample numbers are as in Figure 2).

5) The block-like structures in the selection scan are interpreted as inversions or structural rearrangements that are under selection. Inversions suppress recombination in the inversion, but not in the flanking regions, so if the inversion were under selection, you expect to see the normal decay of signal around the fringes of the inversion. Or is it just that the scale is different so you can't see the decay on the plots? Is it possible that the high differentiation in these regions is due to mapping errors in the rearranged regions? For example, a duplication would lead to apparently highly differentiated allele frequencies.

This pattern is in fact a text-book example of the expected pattern for inversions maintained as balanced polymorphisms over thousands of generations or more (but it is not yet in the text books). The reason is that these are not recent selective sweeps but polymorphisms maintained over long periods of time which allow recombination to randomize the association between the inversion haplotypes and flanking SNPs. We have added text to explain this and cite our previous paper on a 4 MYR old inversion underlying alternative male mating strategies in the ruff that shows exactly the same pattern (Lamichhaney et al., 2016). We can exclude the possibility of a biased estimate of genetic divergence due to presence of duplications within the inverted regions based on normal sequence coverage and based on our PacBio data.

6) A related concern to the first point is whether there are technical batch effects due to i) combination of data sequenced in different experiments (i.e. previously published) and ii) combination of pooled and individual sequencing data. It would be helpful to make sure that these do not further inflate the test statistics.

It is hard to completely rule out batch effects. However, we do not believe that is a problematic issue in this dataset. Our reasons are the following:

i) All contrasts have a very good signal to noise ratio, while batch effects should generate divergence across the genome.

ii) The salinity and spawning time contrasts contain a mix of older and newer samples on either side, yet the results, in both pools and individuals, are highly consistent (see Figure 4—figure supplement 1-3, Figure 5—figure supplement 2)

This view is supported by the results of the PCA analysis based on neutral markers. The tight cluster of herring from Atlantic Ocean (Figure 2A) include herring from all our sequencing batches suggesting that batch effects are not a serious concern in this dataset.

7) Pooling errors. The authors use pools for sequencing and use a Neff allele count correction for read depth to minimise technical bias. They then use the estimated allele count in 2x2 tables to contrast allele frequencies between (super) pools, using an arbitrary p-value threshold of 1e-10. Not observing the actual allele count in the sample can lead to an increased variance and therefore a biased test for association.

While certainly a possibility, we do not think this is a major factor in this case, for reasons outlined in the responses to points 1 and 6. There is further support of the methodology (i. e. linking pool frequencies to individual genotypes using SNP chip analysis) in Martinez-Barrio et al., 2016 and Pettersson et al., 2019.

Furthermore, we have some replicates of presumed same populations that cluster well suggesting that pooling errors is not a serious error in the present study. For instance, samples 2 and 3 represent spring-spawning herring from the Gulf of Riga sampled 2014 and 2016 cluster closely to each other and as a group clusters with other spring-spawning herring from the Baltic Sea sampled during the period 1979 to 2012 (Figure 2—figure supplement 1A). Similarly, samples 4 and 5 represent autumn-spawning herring from the Gulf of Riga sampled 2014 and 2015 and cluster closely and as a group clusters with other spring-spawning herring from the Baltic Sea collected during a period of more than 30-years.

Revisions expected in follow-up work:1) P-value calibration. The selection scan uses an inappropriate null distribution and therefore all the P-values are poorly calibrated and thus uninterpretable. Whichever approach is chosen to calibrate P-values, please assess its performance and include QQ plots in the paper. Without this it's impossible to know how many of the selection hits are actually significant.

We have now provided a set of plots for each contrast that shows the amount of inflation, links *p*-values to DAF and the impact of the threshold on DAF-distribution. In our view, these clearly show that the results are not due to inflation and that the *p*-values are interpretable as a correlate of DAF. We have extended the Materials and methods section to describe these plots.

2) Either remove the claim that the results are in conflict with the infinitesimal model ( our recommendation) or provide a more convincing case to support the claim.

We have modified text and now states that the herring data are strikingly different from the genetic architecture underlying many complex traits in humans, for instance stature. We state that the result is a deviation from expectation under a pure infinitesimal model and we discuss why this is the case. In general, we agree with the reviewer’s views. See point 2 above.

3) Please explain the extent to which the observed differentiation could be due to adaptation, dealing with the points we raised in point 3.

See comments above.

4) Please respond to our criticism of your interpretation of the PCA results.

See our detailed response above.

5) Please defend your interpretation of the block-like structures in the selection scan, in response to our point 5.

Done, see point 5 above.

6) Include batch in the model and check that none of the PCs are predicting batch or data type. Make sure that these do not further inflate the test statistics.

See comments above.

7) Provide data on how pooling errors might have affected your results. Please show Q-Q plots for their statistical test and estimate the sensitivity of your conclusions with respect to the chosen P-value threshold.

See comments above.